# The Minimisation of Cardiovascular Disease Screening for Kidney Transplant Candidates

**DOI:** 10.3390/jcm13040953

**Published:** 2024-02-07

**Authors:** Michael Corr, Amber Orr, Aisling E. Courtney

**Affiliations:** 1Centre for Public Health, Institute of Clinical Sciences B, Royal Victoria Hospital, Queen’s University Belfast, Belfast BT12 6BA, UK; 2Barnsley Hospital NHS Foundation Trust, Barnsley S75 2EP, UK; 3Regional Nephrology & Transplant Unit, Belfast City Hospital, Lisburn Road, Belfast BT9 7AB, UK

**Keywords:** kidney transplantation, cardiovascular disease, screening

## Abstract

**Background**: There is increasing evidence that cardiac screening prior to kidney transplantation does not improve its outcomes. However, risk aversion to perioperative events means that, in practice, testing remains common, limiting the availability of ‘real-world’ data to support any change. Our objective was to assess perioperative and 1-year post-transplant cardiovascular events in a kidney transplant candidate cohort who received minimal cardiovascular screening. **Methods**: The retrospective cohort study included all adult kidney-only transplant recipients in a single UK region between January 2015 and December 2021. Kidney transplant recipients asymptomatic of cardiac disease, even those with established risk factors, did not receive cardiac stress testing. The perioperative and 1-year post-transplant cardiovascular event incidences were examined. Logistic regression was used to identify variables of statistical significance that predicted cardiovascular or cerebrovascular events. **Results**: A total of 895 recipients fulfilled the inclusion criteria. Prior to transplantation, 209 (23%) recipients had an established cardiac diagnosis, and 193 (22%) individuals had a diagnosis of diabetes. A total of 18 (2%) patients had a perioperative event, and there was a 5.7% incidence of cardiovascular events 1 year post-transplantation. The cardiovascular mortality rate was 0.0% perioperatively, 0.2% at 3 months post-transplant, and 0.2% at 1 year post-transplant. **Conclusions**: This study demonstrates comparable rates of cardiovascular events despite reduced screening in asymptomatic recipients. It included higher risk individuals who may, on the basis of screening results, have been excluded from transplantation in other programmes. It provides further evidence that extensive cardiac screening prior to kidney transplantation is unlikely to be offset by reduced rates of cardiovascular events.

## 1. Introduction

Patients with chronic kidney disease have higher rates of coronary heart disease (CHD) compared to the general population [1,2], and this is the leading cause of morbidity and mortality in kidney transplant candidates and recipients [3,4]. Kidney transplant recipients with CHD are at potential greater risk of intraoperative, perioperative, and long-term cardiovascular compromise [4,5]. Observational studies have reported increased cardiovascular events in the perioperative period following transplant, including de novo congestive heart failure and atrial fibrillation [6,7].

The purpose of pretransplant screening for CHD in kidney transplant candidates is to assess left ventricular function [8], identify those who may potentially benefit from preoperative revascularisation (reducing perioperative morbidity and mortality) [9], and identify patients at such an increased risk of perioperative cardiac events that transplantation should not proceed [10]. Within the field of kidney transplantation, there has been increasing controversy over the role of screening for asymptomatic CHD in transplant workup [8]. Despite its generally unverified benefits, many practice guidelines suggest the use of non-invasive cardiac stress testing in asymptomatic or stable CHD kidney transplant candidates [11,12,13,14]. 

There is concern that despite the high rates of CHD in this population, the potential harm and costs of screening outweigh any potential benefits [13,14]. In fact, cumbersome screening processes may even hinder otherwise eligible candidates from receiving a life-saving operation by extending the time until transplant (and the duration of dialysis dependency) or preventing candidates from ever being listed [15]. Moreover, increased cardiac screening increases the cost of the pretransplant preparation of patients, despite currently limited healthcare resources and economic pressures [16]. The results of the ISCHEMIA-CKD (International Study of Comparative Health Effectiveness with Medical and Invasive Approaches–Chronic Kidney Disease) trial, which included kidney transplant candidates, suggests there are no significant benefits to revascularisation in asymptomatic chronic kidney disease patients with normal left ventricular function but positive cardiac stress testing [17,18]. Given the unclear evidence base and controversy, it is unsurprising that there is wide variation in practice across transplant centres in terms of both initial screening and ongoing cardiac surveillance for patients who remain on the transplant waiting list [19,20,21]. 

Since 2015, the Northern Ireland Regional Nephrology and Transplant Centre has adopted a minimisation of the cardiac screening protocol for kidney transplant candidates, removing the requirement of non-invasive functional screening methods for those with no current symptoms of CHD. This study reports the perioperative and 1-year post-transplant cardiovascular and cerebrovascular event incidences in a kidney transplant population with this selected approach. 

## 2. Methods

### 2.1. Study Design

This was a retrospective cohort study using a prospectively designed clinical database—The Northern Ireland Kidney Transplant Database (ethics approval number: REC 23/NI/0034). The database has prospectively collected clinical data for all patients who receive a kidney transplant in this region since its inception in 1961. The data captured include demographic details, past medical history, perioperative and postoperative incidents, graft survival, and patient morbidity and mortality. 

### 2.2. Study Setting

The study was set in the Regional Nephrology and Transplant Unit, Belfast Health and Social Care Trust, Northern Ireland. The centre provides kidney transplantation to the whole of the Northern Irish population (1.9 million), with an average of 115 kidney transplants completed in the centre per annum [22]. The population served is predominantly white and of European ancestry. There is a deceased donor kidney transplant rate of 33.2 per population million (ppm) and a living donor kidney transplant rate of 30 ppm in Northern Ireland. Despite only representing 2.9% of the UK population, Northern Ireland performs ~5.4% of kidney transplants in the UK [22].

Potential kidney transplant recipients are referred by a nephrologist from the unit or from one of four other adult nephrology units throughout the region. A dedicated transplant coordinator nursing team organises the transplant workup according to the agreed protocol and ongoing surveillance once the patient is active on the waiting list. Patients are assessed by a transplant surgeon and formally discussions occur in a transplant multidisciplinary meeting (that includes the referring nephrologist) prior to listing. Deceased donor allocation is managed according to the UK Kidney Offering Scheme. 

At our centre, 53% of patients added to the kidney transplant waiting list are transplanted in less than 1 year, and 42% are subsequently transplanted between 1 and 3 years, with 5% remaining on the waiting list over 3 years [22]. 

### 2.3. Screening Protocols

Prior to 2015, a baseline, then 2-yearly, echocardiogram was mandated for all patients (i) ≥50 years old, (ii) with >2 years spent on dialysis, or (iii) with a past medical history that would indicate an increased risk of CHD (diabetes/ischaemic heart disease/peripheral vascular disease/cerebrovascular disease/impaired left ventricular function/valvular disease/an abnormal electrocardiogram). 

Cardiac stress testing as part of transplant workup, and then 3-yearly, was required for all patients (i) ≥50 years old, (ii) with a history of increased risk of CHD (as above), or (iii) a BMI ≥30 kg/m^2^. 

The cardiac screening currently employed by our unit can be seen in Figure 1. All patients now are assessed annually using the Duke Activity Score Index (DASI) to assess their functional capacity [23]. Echocardiography is not routinely requested, and stress imaging or exercise testing is only performed in symptomatic cardiac disease at the discretion of a cardiologist. All patients prior to listing have a baseline electrocardiogram, which is repeated every 2 years whilst awaiting transplant. Patients with abnormal electrocardiograms are either referred for echocardiograms or for cardiology review. 

### 2.4. Study Population

The study population included all patients aged ≥18 years who had received a kidney transplant between January 2015 and December 2021. Patients who were non-resident in Northern Ireland (typically the Republic of Ireland) were not included due to the limited availability of follow-up data. A comparison population from our centre prior to the minimisation of cardiovascular disease screening was studied between January 2010 and December 2014.

### 2.5. Data Collection

The key demographic data collected from the clinical database included age, gender, and comorbidities. Renal-specific data were also collected, including primary renal disease, renal replacement therapy (RRT) prior to transplantation, RRT start date, and previous transplantation.

Data on the comorbidities associated with an increased risk of CHD, diabetes mellitus, previous cardiac disease diagnosis, peripheral vascular disease, and cerebrovascular disease were collected. The following outcomes were subsequently retrieved and collated. 

Cardiovascular events included admission to a coronary care unit, admission to an intensive care unit for cardiac pathology, myocardial infarction, atrial fibrillation with a fast ventricular response, decompensated heart failure, or mortality secondary to cardiac disease in ≤3 months post-transplantation.Cerebrovascular events included transient ischaemic attack, thrombotic/haemorrhagic stroke, or mortality secondary to stroke in ≤3 months post-transplantation.

The time periods considered were (i) perioperative or early-stage ≤3 months post-transplant and (ii) 1 year post-transplant.

### 2.6. Data Analysis

Statistical analysis was performed using SPSS Statistics version 24 (IBM Corp., Armonk, NY, USA). Logistic regression was used to identify variables of statistical significance that predicted cardiovascular or cerebrovascular events. A *p* value of <0.05 was considered significant.

## 3. Results

A total of 932 individuals received a kidney transplant between January 2015 and December 2021. Ultimately, 895 individuals were included in the study following the removal of paediatric (≤18 years old) and non-resident patients. There were equal numbers of living and deceased donors.

### 3.1. Demographics

The median age at the time of transplant was 53 years (range 18–82), 52% were men, and, in accordance with the local population, individuals were predominantly white. A third of individuals (287) had never required dialysis, and an additional 41 had a pre-emptive re-transplant. The clinical demographics of the studied population can be viewed in Table 1. 

### 3.2. Clinical Risk Factors

Prior to transplantation, 193 (22%) of the total transplant recipients had a diagnosis of diabetes mellitus. The majority of this group, 160 (83%) had end-stage kidney disease due to diabetic nephropathy. A similar proportion, 209 (23%) of recipients, had a formal cardiac diagnosis (excluding left ventricular hypertrophy) prior to transplantation. A smaller minority had previous cerebrovascular (4%) and peripheral vascular (3%) disease. The key comorbidities associated with CHD are outlined in Table 2. 

### 3.3. Perioperative/Early Outcomes

A total of 18 patients had a perioperative or early (≤3 months) cardiac event, representing 2.0% of all kidney transplant recipients. There was a single perioperative death that was not due to cardiac disease and two further deaths within 3 months of transplantation (cardiovascular mortality rate: 0.2%, all-cause mortality rate: 0.3%). The demographic characteristics of the 18 patients who had an early cardiovascular event can be seen in Table 3. 

Perioperatively, two patients required percutaneous coronary intervention (PCI), and four developed atrial fibrillation with a fast ventricular response. Intraoperatively, one patient required management of tachyarrhythmia. Two patients post-operatively were diagnosed with acute stress-induced cardiomyopathy. 

One patient died abruptly and unexpectedly at home seven weeks post-transplantation, and this was presumed to be due to a cardiac event. An additional patient died 11 weeks after transplantation with hypoxia secondary to cardiac dysfunction, following a complex course. Within the 3-month period, three additional patients required PCI (five in total), and four patients were admitted for management of congestive cardiac failure. One patient developed symptomatic bradycardia and required pacemaker placement. No patients had a perioperative or early cerebrovascular event. 

### 3.4. One-Year Outcomes

In the first year after transplantation, one patient was admitted due to a stroke. An additional 33 patients had a cardiac event between three months and one year post-transplantation, representing 3.7% of the total kidney transplant recipients. In addition to earlier complications, this equates to a total 5.7% incidence of cardiovascular events in this cohort in the initial 12 months. 

There were no additional cardiac-related deaths between 3 and 12 months after transplantation; thus, the cardiovascular-related mortality was 0.2% at 1 year post-transplant. 

One patient required a coronary artery bypass graft (CABG), and eight patients required PCI following episodes of unstable angina or myocardial infarction. Three patients had a medically managed acute myocardial infarction, and nine patients were newly diagnosed with atrial fibrillation. Three patients were admitted due to congestive heart failure, and two were diagnosed with a significant valvular defect. Seven patients had cardiac events not associated with CHD, such as pericardial effusion, viral-induced myocarditis, and Takotsubo cardiomyopathy. 

### 3.5. Predictive Factors for Adverse Outcome

Multiple clinical variables were considered potential predictive factors for adverse cardiovascular outcomes after kidney transplantation. Included in a logistic regression model were being male, a deceased kidney donor, dialysis prior to transplantation, a history of diabetes mellitus, previous cardiac history, cerebrovascular history, and the presence of peripheral vascular disease. A separate analysis consisted of perioperative or early (≤3 months) cardiovascular events and 1-year events. 

There were no factors that predicted perioperative/early events. The only clinical variable that was predictive of a cardiovascular event one year post-transplantation was previous cardiovascular disease (OR 1.36, 95% CI 1.05–1.67, *p* < 0.01). The results are shown in Table 4. There was no difference in the median HLA mismatch (3 across A-B-DR) between those patients who did or did not have a cardiac event. There was an insignificant difference in the rates of delayed graft functioning between the groups (*p* = 0.42). The acute rejection rates were low in the whole study population at 7.1%. Two patients with perioperative cardiovascular events had acute cellular rejection, while none had acute antibody-mediated rejection. 

### 3.6. Outcomes Prior to Cardiovascular Disease Screening Minimisation, 2010–2015

The background demographics of the 412 patients who received a kidney transplant at our centre in the five years prior to the minimisation of cardiovascular disease screening (2010–2015) can be seen in Table 5. During this period, two patients died perioperatively/ 3 months post-transplant, one due to pulmonary embolism and one due to presumed sudden cardiac death, giving a cardiovascular mortality rate of 0.26% and an all-cause mortality rate of 0.52%. There was one additional cardiac death between 3 and 12 months post-transplant, equating to a 1-year post-transplant cardiovascular mortality rate of 0.52% for the period January 2010-December 2014. A total of 13 patients had a perioperative or early (≤3 months) cardiac event, representing 3.4% of all kidney transplant recipients. Five patients during this time period had an acute myocardial infarction, five developed a tachyarrhythmia that required medical management, two developed post-operative pulmonary oedema due to poor left ventricular function, and one developed stress-induced myopathy. 

## 4. Discussion

This study involving almost 900 patients demonstrates that minimisation of the cardiac screening of kidney transplant candidates was safe, with low numbers of perioperative and one-year post-transplantation cardiovascular events (2.0% and 5.7%, respectively). The perioperative cardiovascular mortality was 0% and the mortality within 3 months of transplantation was 0.2%. These are consistent with, or lower than, the previously published rates of post-transplant cardiovascular incidents [6,7,24,25] and the rates of cardiovascular events in patients who remain on the waiting list for transplantation [26].

It is noteworthy that these results were achieved in an unselected cohort (no exclusion criteria) with a large proportion of patients who would be considered higher-risk candidates for peritransplant cardiac events (previous cardiac disease 23%, diabetes 22%, re-transplantation 16%). In this population, a past medical history of relevant risk factors was unable to significantly detect those patients who developed a cardiovascular event post-transplantation. One of the two patients who died had no risk factors for cardiac disease, and the other had an historic aortic valve replacement only. Hence, it appears unlikely that enhanced cardiac screening of ‘high-risk’ asymptomatic candidates would have reduced the perioperative morbidity and mortality.

A comparison of the demographics of the patients transplanted between 2010 and 2015 to those transplanted between 2015 and 2020 reveals a picture most transplant centres will recognise. In the more modern era, the transplant recipients are older (median age 53 vs. 48), have greater comorbidity (represented by a higher diabetic nephropathy rate, 18% vs. 8%), and less conservative donor selection (DCD rate, 21% vs. 3%). Despite this, the perioperative cardiac event rate was lower in the years 2015–2020 when screening for cardiovascular disease was minimised (2.0% vs. 3.4% in 2010–2015). The 1-year post-transplant cardiac-related mortality was also lower despite the reduction in cardiovascular disease screening (0.2% vs. 0.52%). Whilst the overall death and event numbers are low in both groups, it is reassuring that a reduction in cardiovascular disease screening was not associated with increased mortality or perioperative cardiac complications. Improvements in modern transplant practices (surgical and medical) may account for the lower number of cardiovascular events in a frailer cohort. It may also demonstrate the more holistic assessment provided by multidisciplinary team meetings (attended by the patients’ own nephrologists), which were introduced at our centre during 2015–2020. 

### 4.1. Limitations

Our study is limited in its generalisability due to being from a single region and overwhelmingly representing a Caucasian population. The heterogeneity in transplant practices may also affect the observed rates of cardiovascular outcomes. The high live donor rate (50%) at our centre may account for reduced perioperative events, though the perioperative cardiac event rate in our deceased donor transplant group of 2.86% also falls below the previously reported studies [24,25,27]. Anaemia is aggressively managed pharmacologically both in the lead-up to transplantation and following the operation. This minimises the ischaemic risk from anaemia and reduces the requirement of blood transfusions and potential fluid overload [28]. Patients presenting for transplantation on beta blockers have these medications continued during the perioperative phase to reduce the risk of rebound tachycardia [29]. Furthermore, our enhanced surgical recovery protocol minimises fluid administration (and potential overload) compared to the fluid prescription practices described in other centres [30,31].

### 4.2. Future Work

Enhanced cardiac screening prior to kidney transplantation has become an embedded part of clinical practice and remains within the guidelines despite a weak evidence base for its efficacy [11,12,13,14]. Furthermore, the results of screening using non-invasive cardiac stress testing has a poor correlation with the occlusive coronary artery disease rates detected using angiography in kidney transplant candidates [17,18]. What is clear is that routine use of enhanced cardiac screening is associated with higher transplant costs and delays to transplantation [14,16]. Most worryingly, patients are inappropriately denied potential transplantation based on unrequired cardiac screening results [15]. 

There is hesitancy to change the clinical approach to pre-transplant cardiac screening despite the observational data demonstrating no benefit [32]. A recent survey of transplant centres in the United Kingdom (UK) displayed overwhelming support for the development of a randomised controlled trial (RCT) in order to change the practice [21]. However, the wait for a landmark clinical trial may be considerable, with preliminary work by Kasiske et al. suggesting that a RCT would need to recruit almost 4000 participants to be adequately powered [33]. 

CARSK (Canadian-Australasian Randomised Trial of Screening Kidney Transplant Candidates for Coronary Artery Disease) is an ongoing multicentre, noninferiority, two-parallel-arm, randomised trial aiming to recruit 3200 kidney transplant candidates to assess whether post-initial listing, periodic cardiac screening whilst on a waiting list has any benefits [34]. There is hope CARSK could be a future template for a large RCT in the field, but it has also demonstrated the significant challenges of any future RCT designed to assess minimised cardiac screening—(1) In the preliminary work related to CARSK, 13/15 surveyed Canadian transplant centres were unwilling to randomise to minimised preliminary cardiac screening in asymptomatic individuals (physician bias). (2) The trial appears to have required expansion beyond the planned initial 26 centres to 39 centres across seven countries and has currently only recruited 1743 of the required 3200 candidates (demonstrating the reality of recruiting such large numbers in a kidney transplant trial) [35]. 

## 5. Conclusions

We suggest that a change in practice should not be delayed while waiting for a large RCT but that, instead, transplant clinicians should initiate a stepwise reduction in unnecessary cardiac screening, accompanied by careful contemporaneous auditing [30]. Our results present such a successful initiative and should encourage others to consider a monitored rationalisation of/reduction in cardiac screening prior to kidney transplantation. In asymptomatic kidney transplant candidates, weak ambiguous evidence of any benefit to cardiac screening and physician anxiety regarding its reduction should not be the rationale for delaying or denying the life-saving intervention that is kidney transplantation. 

## Figures and Tables

**Figure 1 jcm-13-00953-f001:**
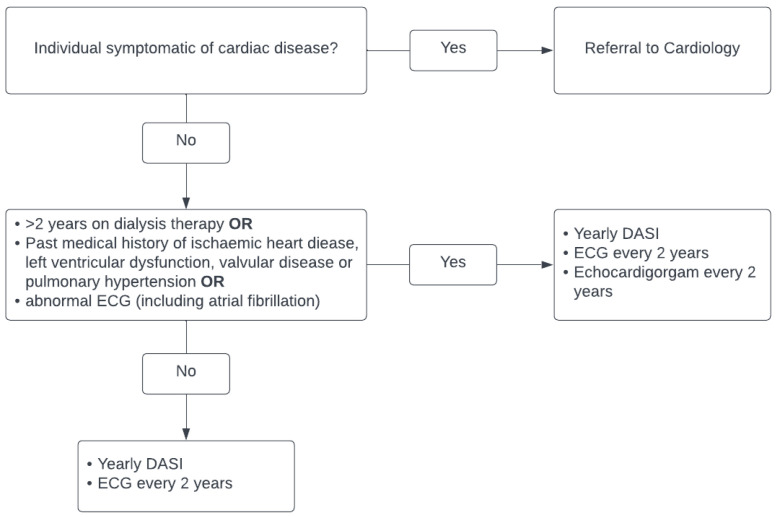
Current cardiac screening for kidney transplant candidates and patients on kidney transplant waiting list in Northern Ireland.

**Table 1 jcm-13-00953-t001:** Clinical characteristics of 895 individuals aged ≥18 years who received a kidney transplant between January 2015 and December 2021.

Clinical Characteristics	Number (% Total KTR Population)
Age (years), median; range	53; 18–82
Gender	
Male	461 (52)
Female	434 (48)
**Primary renal disease**	
Glomerulonephritis	216 (24)
Diabetic nephropathy	160 (18)
Interstitial disease	151 (17)
Other	151 (17)
Unknown	121 (13)
Polycystic kidney disease	96 (11)
**RRT status prior to transplantation**	
Haemodialysis	429 (48)
Pre-emptive, no previous dialysis	287 (32)
Peritoneal dialysis	138 (15)
Low-functioning transplant	41 (5)
**Kidney donor**	
Living	448 (50)
Donation following brain stem death	260 (29)
Donation following circulatory death	187 (21)
Transplant number	
First transplant	748 (84)
Re-transplant	147 (16)

KTR = Kidney transplant recipient.

**Table 2 jcm-13-00953-t002:** Clinical risk factors for coronary heart disease.

Comorbidity	Number
**Diabetes mellitus (total % KTR population)**	**193 (22%)**
Type 1 diabetes mellitus	59
Type 2 diabetes mellitus (insulin-controlled)	83
Type 2 diabetes mellitus (medication-controlled)	31
Type 2 diabetes mellitus (diet-controlled)	20
**Previous cardiac disease ^1^ (total % KTR)**	**209 (23%)**
Valvular disease	69
Previous PCI/CABG	55
Atrial fibrillation	45
Other	44
Congestive cardiac failure	29
Medically managed ischaemic heart disease	20
**Previous cerebrovascular disease (total % KTR)**	**39 (4%)**
Stroke	28
Transient ischaemic attack	11
**Peripheral vascular disease ^2^ (total % KTR)**	**26 (3%)**

KTR = kidney transplant recipient. PCI = percutaneous coronary intervention. CABG = coronary artery bypass graft. ^1^ A total of 38 patients had >1 cardiovascular comorbidity. ^2^ Defined as patients with peripheral vascular disease requiring percutaneous or vascular surgical intervention.

**Table 3 jcm-13-00953-t003:** Clinical characteristics of 18 individuals who had a perioperative or early (≤3 months post-transplantation) cardiovascular event.

Clinical Characteristics	Number
Age (years), median; range	53 (44–76)
Gender	
Male	11
Female	7
**Comorbidity ^1^**	
Previous cardiovascular disease	9
Diabetes mellitus	8
Previous cerebrovascular disease	3
Previous peripheral vascular disease	0
**RRT status prior to transplantation**	
Haemodialysis	10
Pre-emptive, no previous dialysis	5
Peritoneal dialysis	3
Low-functioning transplant	0
**Kidney donor**	
Living	6
Donation following brain stem death	5
Donation following circulatory death	7
**Transplant number**	
1st transplant	18
Re-transplant	0

^1^ Two patients had two comorbidities.

**Table 4 jcm-13-00953-t004:** Logistic regression analysis of predictors of cardiovascular event 1 year post-transplantation.

Clinical Variable	Odds Ratio	95% CI	*p* Value
Male Gender	0.03	−0.28–0.34	0.86
Deceased Donor	0.49	0.17–0.81	0.12
Dialysis Pre-Transplant	0.14	−0.2–0.48	0.67
Diabetes Mellitus	1.29	0.96–1.62	0.48
Cardiovascular Disease	1.36	1.05–1.67	<0.01
Cerebrovascular Disease	0.4	−0.15–1.01	0.46
Peripheral Vascular Disease	1.51	0.96–2.06	0.25

**Table 5 jcm-13-00953-t005:** Clinical characteristics of 388 individuals aged ≥18 years who received a kidney transplant between January 2010 and December 2014.

Clinical Characteristics	Number (% Total KTR Population)
Age (years), median; range	48; 18–77
Gender	
Male	249 (64)
Female	139 (36)
**Primary renal disease**	
Glomerulonephritis	97 (25)
Interstitial disease	85 (22)
Other	77 (20)
Polycystic kidney disease	66 (17)
Unknown	32 (8)
Diabetic nephropathy	31 (8)
**RRT status prior to transplantation**	
Haemodialysis	232 (60)
Pre-emptive, no previous dialysis	78 (20)
Peritoneal dialysis	70 (18)
Low-functioning transplant	8 (2)
**Kidney donor**	
Living	237 (61)
Donation following brain stem death	139 (36)
Donation following circulatory death	12 (3)
**Transplant number**	
First transplant	311 (80)
Re-transplant	77 (20)

KTR = kidney transplant recipient.

## Data Availability

Anonymized data available upon request to corresponding author. Not publicly shared as a clinical information contained.

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
