# Peer review of "The Minimisation of Cardiovascular Disease Screening for Kidney Transplant Candidates"

_jcm, 2024, doi:10.3390/jcm13040953_

Round 1

Reviewer 1 Report

Comments and Suggestions for Authors

The study has an interesting point of view on a debated topic. A long story of a still hot topic - until we have enough scientific resources from RCTs to rely on.

It is out of question that we need to optimize cardiac screening of kidney transplant candidates, however I think this study's usefulness has to be valued more in a regional context (namely Northern Ireland), than it can be generally applied to kidney transplant candidate cohorts. Major point: high  proportion (50%) of live donors (beautiful!), which is thought to reduce rates of cardiovascular events for the whole population. These lower rates were compared to data from literature. 

The authors choose cardiac symptomatics as the decisive point of referral to cardiology. If there are no such symptoms, some (but not all!) of known major cardiovascular risk factors are used to screen patients for indicating echocardiography.

Although logistic regression analysis of predictors for cardiovascular event only showed previous cardiovascular disease as a risk factor (1 year posttransplant), one might expect that with higher number of transplantations some other factors might have also significance: e.g. diabetes, or dialysis pretransplant.

Questions and concerns:

1. Rates from a regional cohort of patients with high proportion of living donations were compared to historic data. Were the proportions of living donations in those historic studies comparable?

2. Diabetes is not mentioned as a risk factor to be screened for when there are no symptoms of cardiac diseases. Why? In case of silent ischemia (more frequent in DKD) patients might not present symptoms of cardiac disease, and these patients might not be filtered out in time by the authors' screening system. Please comment.

3. I miss hypertension among causes of primary renal disease.

4. We have no information on smoking status.

5. As we do not have the RCTs yet that would prove that such a minimisation of screening is the way to go, I would be more cautious to use successful minimisation in the title. E.g.: "Minimisation of Cardiovascular Disease Screening for Kidney Transplant Candidates Who Do Not Present Cardiac Symptoms." or similar.

Author Response

The study has an interesting point of view on a debated topic. A long story of a still hot topic - until we have enough scientific resources from RCTs to rely on. It is out of question that we need to optimize cardiac screening of kidney transplant candidates, however I think this study's usefulness has to be valued more in a regional context (namely Northern Ireland), than it can be generally applied to kidney transplant candidate cohorts. Major point: high  proportion (50%) of live donors (beautiful!), which is thought to reduce rates of cardiovascular events for the whole population. These lower rates were compared to data from literature.  The authors choose cardiac symptomatics as the decisive point of referral to cardiology. If there are no such symptoms, some (but not all!) of known major cardiovascular risk factors are used to screen patients for indicating echocardiography. Although logistic regression analysis of predictors for cardiovascular event only showed previous cardiovascular disease as a risk factor (1 year posttransplant), one might expect that with higher number of transplantations some other factors might have also significance: e.g. diabetes, or dialysis pretransplant.

We thank the reviewer for their kind comments. In our discussion we outline the limitations of a study coming from a single region. However, with vast heterogeneity in the approach clinically to this problem it may be until an RTC is performed (the challenges of which we also discuss) that we rely on observational evidence from individual regions practicing in a similar manner.  We agree that very large studies may unmask other factors of significance, however the absence of such associations in a cohort of almost 900 patients we feel is noteworthy.

Questions and concerns:

  1. Rates from a regional cohort of patients with high proportion of living donations were compared to historic data. Were the proportions of living donations in those historic studies comparable?

As we outline in our discussion section the high living donor rates may explain our relatively low event rate, however even our cardiac episode event rate in deceased donors is lower than other reported studies. The following has been added to our discussion.

“The high live donor rate (50%) at our centre may account for reduced perioperative events, though the perioperative cardiac event rate of our deceased donor transplant group of 2.86% also falls below previous reported studies”

  1. Diabetes is not mentioned as a risk factor to be screened for when there are no symptoms of cardiac diseases. Why? In case of silent ischemia (more frequent in DKD) patients might not present symptoms of cardiac disease, and these patients might not be filtered out in time by the authors' screening system. Please comment.

Whilst the reviewers assertions about true asymptomatic cardiac disease in patients with diabetes are correct, as outlined in the “2023 European Society of Cardiology Guidelines for the management of cardiovascular disease in patients with Diabetes” screening for coronoary artery disease (CAD) disease in diabetic patients is also controversial (1). “Various RCTs evaluating the impact of routine screening for CAD in asymptomatic patients with diabetes and no history of CAD showed no differences in CV outcomes at follow-up in those who underwent routine screening compared with standard recommendations.” As outlined in our manuscript all patients undergo ECG assessment, where a significant historic ischaemic event would be detected leading to the patient receiving an echocardiogram for assessment. The ACC/AHA guidance also reflect that no controlled studies show an advantage for angiography or revascularisation in the diabetes subgroup (2). To clarify this further we have added in the following statement, “All patients prior to listing have a baseline electrocardiogram, which is repeated every 2 years whilst awaiting transplant. Patients with abnormal electrocardiograms are either referred for echocardiogram or for cardiology review.”

  • Nikolaus Marx, Massimo Federici, Katharina Schütt, et al , 2023 ESC Guidelines for the management of cardiovascular disease in patients with diabetes: Developed by the task force on the management of cardiovascular disease in patients with diabetes of the European Society of Cardiology (ESC), European Heart Journal, Volume 44, Issue 39, 14 October 2023, Pages 4043-4140, https://doi.org/10.1093/eurheartj/ehad192
  • Writing Committee Members; Lawton JS, Tamis-Holland JE, Bangalore S, Bates ER, Beckie TM, Bischoff JM, Bittl JA, Cohen MG, DiMaio JM, Don CW, Fremes SE, Gaudino MF, Goldberger ZD, Grant MC, Jaswal JB, Kurlansky PA, Mehran R, Metkus TS Jr, Nnacheta LC, Rao SV, Sellke FW, Sharma G, Yong CM, Zwischenberger BA. 2021 ACC/AHA/SCAI Guideline for Coronary Artery Revascularization: A Report of the American College of Cardiology/American Heart Association Joint Committee on Clinical Practice Guidelines. J Am Coll Cardiol. 2022 Jan 18;79(2):e21-e129. doi: 10.1016/j.jacc.2021.09.006.

  • I miss hypertension among causes of primary renal disease.

Our transplant (and general) population is overwhelmingly Caucasian. Hence as a region we treat hypertensive nephropathy as a disease of exclusion. As emphasised in this article published by the European Renal Registry (1) “hypertension-induced end stage renal disease might not exist at all..” despite its widescale reporting. Clinically, as a region, we work with pathology and genetics colleagues to identify rare causes and cases of familial nephropathy. In the UK initiative “100 000 Genome project” (2) we were the highest contributing specialty in Northern Ireland. This has minimised in recent years the number of patients being coded as “hypertensive nephrosclerosis” as their primary renal disease. Those remaining are included in the “other category”, NHSBT data would suggest ~4% of our patients attending for transplant are coded as hypertensive nephropathy.  

  • Carriazo S, Vanessa Perez-Gomez M, Ortiz A. Hypertensive nephropathy: a major roadblock hindering the advance of precision nephrology. Clin Kidney J. 2020 Sep 2;13(4):504-509. doi: 10.1093/ckj/sfaa162. PMID: 32897275; PMCID: PMC7467619.
  • Kerr K, McKenna C, Heggarty S, Bailie C, McMullan J, Crowe A, Kilner J, Donnelly M, Boyle S, Rea G, Flanagan C, McKee S, McKnight AJ. A Formative Study of the Implementation of Whole Genome Sequencing in Northern Ireland. Genes (Basel). 2022 Jun 21;13(7):1104. doi: 10.3390/genes13071104. PMID: 35885887; PMCID: PMC9316942.
  1. We have no information on smoking status.

Unfortunately smoking status is not collected on the Northern Ireland Kidney Transplant Database so we are unable to provide this information.

  1. As we do not have the RCTs yet that would prove that such a minimisation of screening is the way to go, I would be more cautious to use successful minimisation in the title. E.g.: "Minimisation of Cardiovascular Disease Screening for Kidney Transplant Candidates Who Do Not Present Cardiac Symptoms." or similar.

Have changed title based on the reviewers suggestion.

Reviewer 2 Report

Comments and Suggestions for Authors

This study examines an important, topical, and clinically relevant question in kidney transplantation. The study is well-designed and well-presented. Their conclusions are supported by the data and the authors do not over-reach in their interpretation. Study limitations which restrict widespread generalisability are acknowledged appropriately. This study marks a valuable contribution to the 'real world' data  which suggests that the current widespread practise of extensive cardiovascular screening could successfully be tapered following appropriate clinical stratification without adverse clinical consequences, and with the potential benefit of reducing transplant waitlist time.  I congratulate the authors on this work. 

I would suggest a careful re-read for a small number of typographical, formatting, and occasional grammatical errors examples of which include the following:

Line 24: 'stud were'  ( instead of 'study where')

Line 58/59: 'the potential harm and costs outweighs any potential benefits'

Line 134:' preoperative or early (<3 months) of transplant ' ....(either remove brackets or change to post-transplant)

Line 263: reference 34 formatted incorrectly

Author Response

This study examines an important, topical, and clinically relevant question in kidney transplantation. The study is well-designed and well-presented. Their conclusions are supported by the data and the authors do not over-reach in their interpretation. Study limitations which restrict widespread generalisability are acknowledged appropriately. This study marks a valuable contribution to the 'real world' data  which suggests that the current widespread practise of extensive cardiovascular screening could successfully be tapered following appropriate clinical stratification without adverse clinical consequences, and with the potential benefit of reducing transplant waitlist time.  I congratulate the authors on this work. 

We thank the reviewer for their kind comments.

I would suggest a careful re-read for a small number of typographical, formatting, and occasional grammatical errors examples of which include the following:

Line 24: 'stud were'  ( instead of 'study where')

Changed to “study” as suggested thanks for spotting

Line 58/59: 'the potential harm and costs outweighs any potential benefits'

Changed as suggested.

Line 134:' preoperative or early (<3 months) of transplant ' ....(either remove brackets or change to post-transplant)

Changed to “post-transplant” as suggested

Line 263: reference 34 formatted incorrectly

Reformatted

Reviewer 3 Report

Comments and Suggestions for Authors

To the authors,

Thank you for the opportunity to review this article.

In this study, the authors investigated the peri-operative and 1-year post transplant cardiovascular events in a kidney transplant candidate cohort who received minimal cardiovascular screening. And the authors showed comparable rates of cardiovascular events despite reduced screening in asymptomatic recipients.

This study provides important insight in screening candidates for kidney transplant. However, this study only includes kidney transplant candidates after 2015, when screening was simplified. Detailed comparisons with pre-2015 patients will be important to determine if screening minimization was truly successful. It is important to compare not only mortality, but also patient background and cost. I consider this a critical issue.

Author Response

Thank you for the opportunity to review this article. In this study, the authors investigated the peri-operative and 1-year post transplant cardiovascular events in a kidney transplant candidate cohort who received minimal cardiovascular screening. And the authors showed comparable rates of cardiovascular events despite reduced screening in asymptomatic recipients.

No comment felt required.

This study provides important insight in screening candidates for kidney transplant. However, this study only includes kidney transplant candidates after 2015, when screening was simplified. Detailed comparisons with pre-2015 patients will be important to determine if screening minimization was truly successful. It is important to compare not only mortality, but also patient background and cost. I consider this a critical issue.

We cannot give cost analysis for any study period as this data is not available to us nor was it the aim of our research. We have provided in a new section of our results (section 3.6) comparative demographics from the 5 years prior to the minimization of cardiovascular disease screening. 3 month and 1-year post-transplant mortality and perioperative cardiac events for this period. Some data we do not have available such as pre-transplant diabetes rate. However, we feel the fact that the 2010-2015 were a younger cohort (median age 48 vs 53), had less diabetic nephropathy (8% vs 18%), a higher live donor rate (60% vs 50%) can demonstrate that we (like all centres) are transplanting older more comorbid patients in the modern period. Despite this and the reduction in cardiac screening our cardiovascular mortality and cardiac event rate perioperatively was lower in 2015-2020. We feel this allows the reader to be adequately reassured that reduced cardiac screening has not been associated with a rise in cardiovascular death or events.

“The background demographics of the 412 patients who received a kidney transplant at our centre in the 5-years prior to the minimisation of cardiovascular disease screening (2010-2015) can be seen in Table 5. During this period two patients died perioperatively/ 3 months post-transplant, one due to pulmonary embolism and one due to presumed sudden cardiac death; giving a cardiovascular mortality rate of 0.26% and an all-cause mortality rate of 0.52%. There was one additional cardiac death between 3-12 months post-transplant equating to a 1-year post-transplant cardiovascular mortality rate of 0.52% for the period January 2010-December 2014. A total of 13 patients had a perioperative or early (≤ 3 months) cardiac event representing 3.4% of all kidney transplant recipients. Five patients during this time period had an acute myocardial infarction, five developed a tachyarrhythmia that required medical management, two developed post-operative pulmonary oedema due to poor left ventricular function and one developed a stress induced myopathy.”

We have added further analysis to this data in the discussion section;

Comparison of demographics of patients transplanted between 2010-2015 to those transplanted between 2015-2020 reveal a picture most transplant centres will recognise. In the more modern era transplant recipients are older (median age 53 vs. 48), have more comorbidity (represented by higher diabetic nephropathy rate 18% vs. 8%) and donor selection less conservative (DCD rate 21% vs. 3%). Despite this, the perioperative cardiac event rate was lower in the years 2015-2020 when screening for cardiovascular disease was minimised (2.0% vs. 3.4% in 2010-2015).  1-year post-transplant cardiac-related mortality was also lower despite the reduction in cardiovascular disease screening (0.2% vs. 0.52%). Whilst overall death and event numbers are low in both groups, it is reassuring that a reduction in cardiovascular disease screening was not associated with increase mortality of perioperative cardiac complications. Improvements in modern transplant practices (surgical and medical) may account for lower cardiovascular events in a frailer cohort. It may also demonstrate the more holistic assessment provided by a multidisciplinary team meeting (attended by the patient’s own nephrologist) which was introduced at our centre during 2015-2020.”

Reviewer 4 Report

Comments and Suggestions for Authors

Corr et all present interesting results from their experience with minimizing cardiovascular disease screening in a cohort of patients who underwent kidney transplantation at their center between January 2017 and December 2021. They reported out of 895 patients in the study (50% live donor, 29% donation after brain death and 21% donation after cardiovascular death) that there were no perioperative cardiovascular deaths, and a death rate of 0.2% at three months and 0.2% at 1-year post transplant. They note that generalizability of their results is limited since the bulk of patients were white Northern Europeans and because the study was single center. They also note the high percentage of living donors. Strengths include that there were no exclusion criteria beyond participants less than 18 years old or less and non-residents of Northern Ireland. Cardiovascular events at 3 months were 2% and 5.7% at one year.

The article focuses on the recipients but mentions nothing about donors beyond whether they were living donors, DBD or DCD donors. Some mention of the quality of the kidneys accepted for transplant should be made. Was there any association in the few deaths with delayed graft function, prolonged warm or cold ischemia time, age and comorbidities of the donors? How much selectivity was there in matching donors and recipients in terms of expected survival of organ and recipient? Was there any association with the degree of HLA mismatches? Was there any association with early rejection, either cellular or antibody mediated?

These may be questions that only a large study can answer, but should still be mentioned to allow readers to assess whether results are applicable to their patients and donor pool.

The authors note the concerns that remaining on dialysis poses risk and suggest the risks of being turned down for a kidney as a result of cardiac testing. How does their death rate among recipients compare to death rate on the waiting list for first time transplants and for re-transplants. What is the average waiting time for a kidney at their center and what is the catchment area for donors at their center?

Does their center do any testing for exercise tolerance - exercise stress tests, cardiopulmonary metabolic testing to assess a candidates suitability and exclude potential recipients with low functional capacity? Do the authors conclude that this testing is also unnecessary?

Finally, what is the possibility of selection bias in terms of referring physicians sending patients to the transplant center for evaluation? Could it be that patients considered at high risk of cardiovascular events were never referred?

Author Response

Corr et all present interesting results from their experience with minimizing cardiovascular disease screening in a cohort of patients who underwent kidney transplantation at their center between January 2017 and December 2021. They reported out of 895 patients in the study (50% live donor, 29% donation after brain death and 21% donation after cardiovascular death) that there were no perioperative cardiovascular deaths, and a death rate of 0.2% at three months and 0.2% at 1-year post transplant. They note that generalizability of their results is limited since the bulk of patients were white Northern Europeans and because the study was single center. They also note the high percentage of living donors. Strengths include that there were no exclusion criteria beyond participants less than 18 years old or less and non-residents of Northern Ireland. Cardiovascular events at 3 months were 2% and 5.7% at one year.

We thank the reviewer for their kind comments.

The article focuses on the recipients but mentions nothing about donors beyond whether they were living donors, DBD or DCD donors. Some mention of the quality of the kidneys accepted for transplant should be made. Was there any association in the few deaths with delayed graft function, prolonged warm or cold ischemia time, age and comorbidities of the donors? How much selectivity was there in matching donors and recipients in terms of expected survival of organ and recipient? Was there any association with the degree of HLA mismatches? Was there any association with early rejection, either cellular or antibody mediated? These may be questions that only a large study can answer but should still be mentioned to allow readers to assess whether results are applicable to their patients and donor pool.

With regards to kidney allocation, deceased donor organs in the UK are considered a national resource with clear allocation policies to ensure equity of access to, and utility of organs is maximised. Northern Ireland is part of the UK and receive kidney offers according to the allocation algorithm. We have clarified this in section 2.2 with “Deceased donor allocation is managed according to the UK Kidney Offering Scheme.” This should give the reader an interpretation of the way organs are allocated to individual recipients.

We have included the following information in section 3.5 regarding predictive factors for adverse outcomes,

There was no difference in the median HLA mismatch (3 across A-B-DR) between those patients who did or did not have a cardiac event. There was an insignificant difference in the rates of delayed graft functioning between the groups (p=0.42). Acute rejection rates were low in the whole study population at 7.1%. Two patients with perioperative cardiovascular events had acute cellular rejection, none had acute antibody mediated rejection.”

As the reviewer has alluded to the study is too small (and cases of acute rejection too low) to really read into these figures so we have displayed them for contextual purposes as suggested.

The authors note the concerns that remaining on dialysis poses risk and suggest the risks of being turned down for a kidney as a result of cardiac testing. How does their death rate among recipients compare to death rate on the waiting list for first time transplants and for re-transplants.

Unfortunately, we do not have access to any local data regarding this as our ethics only covers patients who have received a kidney transplant. However, we have referenced recent studies in our discussion around mortality of patients remaining on the waiting list for kidney transplantation (which is much higher). Reference below.

Chen HH, Chern YB, Hsu CY, Tang PL, Lai CC. Kidney transplantation waiting times and risk of cardiovascular events and mortality: a retrospective observational cohort study in Taiwan. BMJ Open. 2022 May 24;12(5):e058033. doi: 10.1136/bmjopen-2021-058033.

What is the average waiting time for a kidney at their center and what is the catchment area for donors at their center?

We agree with the reviewer this is important contextual information, we have in our previous comments outlined we are part of the UK allocation scheme. We have also added the following to section 2.2.

At our centre, 53% of patients added to the kidney transplant waiting list are transplanted in less than 1 year. 42% are subsequently transplanted between 1-3 years, with 5% remaining on the waiting list over 3 years.”

Does their center do any testing for exercise tolerance - exercise stress tests, cardiopulmonary metabolic testing to assess a candidates suitability and exclude potential recipients with low functional capacity? Do the authors conclude that this testing is also unnecessary?

As outlined in section 2.3 of our manuscript all transplant recipients are assessed by Duke Activity Score Index to assess functional capacity. To clarify this further in our manuscript we have modified the following paragraph,

The cardiac screening currently employed by our unit can be seen in Figure 1. All patients now are assessed annually with Duke Activity Score Index (DASI) to assess functional capacity, 23 echocardiography is not routinely requested, and stress imaging or exercise testing is only performed in symptomatic cardiac disease at the discretion of a cardiologist. All patients prior to listing have a baseline electrocardiogram, which is repeated every 2 years whilst awaiting transplant. Patients with abnormal electrocardiograms are either referred for echocardiogram or for cardiology review.”

Finally, what is the possibility of selection bias in terms of referring physicians sending patients to the transplant center for evaluation? Could it be that patients considered at high risk of cardiovascular events were never referred?

Theoretically this could be the case, and very hard to disprove! But given the very high kidney transplant rates in Northern Ireland it would seem unlikely. In section 2.2 we have added further details regarding transplant rates in Northern Ireland.

“There is a deceased donor kidney transplant rate of 33.2 per population million (ppm) and a living donor kidney transplant rate of 30ppm in Northern Ireland. Despite only representing 2.9% of the UK population, Northern Ireland performs ~5.4% of the kidney transplants in the UK.”

We also feel having outlined the baseline population characteristics of our population 22% with diabetes, 23% with previous cardiac disease, 4% with previous cerebrovascular disease and 3% with previous peripheral vascular disease (Table 2) allows the reader to reflect on the comorbidity of the population we have included in the study and compare with their own transplant population.

Round 2

Reviewer 3 Report

Comments and Suggestions for Authors

To the authors,

Thank you for the opportunity to review the revised manuscript and for taking into consideration my suggestions. This study was dramatically improved, and I find the revised version much more suitable for publication.